# Selective O_2_/N_2_ Separation Using Grazyne Membranes: A Computational Approach Combining Density Functional Theory and Molecular Dynamics

**DOI:** 10.3390/nano14242053

**Published:** 2024-12-22

**Authors:** Adrià Calzada, Francesc Viñes, Pablo Gamallo

**Affiliations:** Departament de Ciència de Materials i Química Física, Institut de Química Teòrica i Computacional (IQTCUB), Universitat de Barcelona, C/Martí i Franquès 1-11, 08028 Barcelona, Spain; adria.calzada@ub.edu (A.C.); francesc.vines@ub.edu (F.V.)

**Keywords:** grazynes, density functional theory, molecular dynamics, membranes, gas separation, kinetics

## Abstract

The separation of oxygen (O_2_) and nitrogen (N_2_) from air is a process of utmost importance nowadays, as both species are vital for numerous fundamental processes essential for our development. Membranes designed for their selective molecule separation have become the materials of choice for researchers, primarily due to their ease of use. The present study proposes grazynes, 2D carbon-based materials consisting of *sp* and *sp*^2^ C atoms, as suitable membranes for separating O_2_ and N_2_ from air. By combining static density functional theory (DFT) calculations with molecular dynamics (MD) simulations, we address this issue through a comprehensive examination of the thermodynamic, kinetic, and dynamic aspects of the molecular diffusions across the nano-engineered pores of grazynes. The studied grazyne structures have demonstrated the ability to physisorb both O_2_ and N_2_, preventing material saturation, with diffusion rates exceeding 1 s^−1^ across a temperature range of 100–500 K. Moreover, they exhibit a selectivity of *ca.* 2 towards O_2_ at 300 K. Indeed, MD simulations with equimolar mixtures of O_2_:N_2_ indicated a selectivity towards O_2_ in both grazynes with *ca*. twice as many O_2_ filtered molecules in the [1],[2]{2}-grazyne and with O_2_ representing ca. 88% of the filtered gas in the [1],[2]{(0,0),2}-grazyne. [1],[2]{2}-grazyne shows higher permeability for both molecules compared to the other grazyne, with O₂ demonstrating particularly enhanced diffusion capacity across both membranes. Further MD simulations incorporating CO_2_ and Ar confirm O_2_ enrichment, particularly with [1],[2]{(0,0),2}-grazyne, which increased the presence of O_2_ in the filtered mixture by 26% with no evidence of CO_2_ molecules.

## 1. Introduction

The Earth’s atmosphere is made up of a number of different gases, which form a perfect mixture so that life on our planet has been able to develop for millions of years. Among all the present gases, there are two that stand out above the rest: oxygen (O_2_) and nitrogen (N_2_), both essential for the existence of life on Earth. N_2_ represents 78% of the composition of the atmosphere [1], and, without it, life as we know it could not be understood due to the so-called nitrogen cycle, where N_2_ moves between the atmosphere, soil, living beings, and water, thus nourishing plants and regulating the food chain. On the other hand, O_2_ represents 21% of the gases present in the atmosphere [1], and, like N_2_, is one of the most important chemical elements since it is present, directly or indirectly, in almost all known biological functions. The combination of these two compounds forms a perfect mixture for the survival of living beings but, sometimes, one needs to separate them to be able to use them in different processes of technological importance. Thus, N₂ is used, for instance, in food and beverage packaging as a colorless, inert, and odorless compound in the so-called modified atmosphere packaging (MAP) [2], which reduces the oxidation of the product as well as the growth of aerobic microbes [3]. Due to its inert nature, N_2_ is commonly used in chemistry to create an inert atmosphere, especially for those reactions involving highly reactive compounds [4]. Furthermore, N_2_ also plays a crucial role in processes like the Haber–Bosch (HB) process, where atmospheric N_2_ is used as a reactant under high pressure and temperature to produce ammonia (NH_3_), which is subsequently used, along with other compounds such as carbon dioxide (CO_2_) or sulfuric acid (H_2_SO_4_), in the production of nitrogen fertilizers. As for O_2_, it is also used in MAP [2], but its applications go far beyond this. For instance, the steel industry is the largest consumer of pure O_2_, using it in the process of blowing high-carbon steel [5]. In the chemical industry, commercial O_2_ or oxygen-enriched air is used in the synthesis of controlled oxidation products [5]. Additionally, O_2_ has medical uses, such as in oxygen therapy, which is used to treat any condition that affects the body ability to obtain and use oxygen [6].

Even though the separation of these gases—by the so-called chemical resolution—might seem simple, this statement could not be further from truth. The high stability and concomitant low reactivity of both molecules makes it difficult to separate them. In recent years, the use of different types of membranes has become a suited solution to this problem. Different types of membranes exist, predominantly categorized into organic and inorganic families. Organic membranes are principally constituted of polymers [7], whereas inorganic membranes comprise materials such as carbon and ceramics, which are mainly made of Al, Si, Ti, or Zn [8]. This latter group of membranes has been shown to be thermally stable and chemically resistant, but with very high production costs in full-scale processes [9]. Over the recent years, carbon-based membranes have gained significant attention, particularly after the discovery of graphene [10]. Membranes composed of porous materials have emerged as candidates for molecular separation, taking advantage of the difference in diffusion energies to selectively facilitate some given species permeation. Noteworthy examples include graphene oxide (GO) membranes, which have demonstrated high selectivity in separating various gas mixtures [11]. Encouraging results have also been observed on graphene nanostructures and reduced GO (rGO) applied in biogas upgrading processes [12,13]. Additionally, nanoporous graphene has exhibited notable efficiency in the separation of gas mixtures such as H_2_/N_2_, owing to their exceptional pore tunability [14].

Another class of carbon-based materials exhibiting favorable results are graphynes [15,16], single-sheet C-based materials like graphene, but characterized by combining C atoms with *sp-* and *sp*^2^-hybridizations in an ordered fashion. These structures are proposed as gas separation membranes due to their structural versatility, enabling the nanoengineering of their pores at will. The study of graphynes has given rise to a novel structural variant named grazynes [17], which comprise graphene stripes with *sp*^2^ hybridization linked by acetylenic linkages. From an electronic perspective, grazyne materials exhibit novel electronic transport that occurs perpendicular to the acetylene bonds [18]. Besides, the grazyne family is also highly tunable, similarly to graphynes, either through elongation of acetylene units or graphene stripes, or by creating vacancies on the acetylene linkers. In fact, the first investigations into the application of grazynes as separation materials have yielded promising results in biogas upgrading [19] and in the separation of N_2_/CO_2_ from air mixtures [20].

The present study aims at exploring, by computational methods, whether the separation of N_2_/O_2_ gases mixtures is effectively suitable when using adapted grazyne structures as separation membranes. This study, carried out in a comprehensive, multiscale way, encompasses the examination of thermodynamic, kinetic, and dynamic aspects demonstrating the ability of both O_2_ and N_2_ surfaces to physisorb, with diffusion rates exceeding 1 s^−1^ across a temperature range of 100–500 K. Moreover, they exhibit a selectivity of ca. 2 towards O_2_ at 300 K. Molecular dynamics (MD) simulations with equimolar mixtures of O_2_:N_2_ indicated a selectivity towards O_2_ in both grazynes with ca. twice as many O_2_ filtered molecules. Further MD simulations incorporating CO_2_ and Ar confirm O_2_ enrichment, particularly with [1],[2]{(0,0),2}-grazyne, which increased the presence of O₂ in the filtered mixture by 26%, with no evidence of CO_2_ molecules.

## 2. Computational Details

This study is based on computational simulations employing periodic density functional theory (DFT) calculations with the Vienna *ab initio* simulation package (VASP) [21]. The core electron density was described by the projector augmented wave (PAW) method [22], while the valence electron density expanded on a planewave basis set with a kinetic energy cutoff of 415 eV. To address exchange-correlation (*xc*) effects, the Perdew–Burke–Ernzerhof (PBE) *xc* functional was utilized [23]. Additionally, Grimme’s D3 description of dispersive forces (PBE-D3) [24] has been added to precisely capture dispersion interactions. Isolated O_2_ and N_2_ molecules were fully optimized at **Γ k**-point within a large cubic unit cell with dimensions of 10 × 10 × 10 Å^3^, setting electronic and ionic convergence thresholds at 10^−6^ eV and 10^−5^ eV, respectively.

The investigated grazynes, named as [1],[2]{2}-grazyne and [1],[2]{(0,0),2}-grazyne, were chosen based on their favorable outcomes in biogas upgrading [19]. These grazynes are made with one-dimensional graphene stripes linked to each other by pairs of acetylenic bonds. Both structures feature a pore characterized by two consecutive acetylenic vacancies; see Figure 1 and Figure 2. The main difference lies in the arrangement of acetylenic bonds within the pore region; the [1],[2]{2}-grazyne contains a single acetylenic row between consecutive pore units, while the [1],[2]{(0,0),2}-grazyne exhibits two adjacent acetylenic rows followed by two vacancies, as indicated by the {(0,0),2} notation. For any clarification on the nomenclature of this type of materials, we refer to the literature where it is explained in detail [17].

These grazynes were fully optimized after introducing a perpendicular vacuum region of 10 Å above and below the membrane. This configuration aimed to prevent interactions between periodically repeated layers of grazyne. A Monkhorst–Pack **k**-point grid of 10 × 10 × 1 was used to ensure energy convergence below the chemical accuracy threshold of 1 kcal·mol^−1^, *ca*. 0.04 eV. The optimized cell parameters for each grazyne model can be found in Table 1. The pore sizes in both grazyne structures were estimated using different methods resulting in pore areas of 59.27 Å^2^ using C atoms as single points, a value that reduces into 44.27 Å^2^ when subtracting the C atomic covalent radius and 52.57 Å^2^ considering Bader’s procedure [20].

The determination of diffusion transition states (TS) corresponding to the passage of the molecules across the pore was systematically conducted. Initially, the center of mass of both N_2_ and O_2_ molecules was aligned with the geometric center of the pore and positioned *ca*. 4 Å above the layer, and oriented either perpendicular or parallel to the grazyne plane. Throughout the optimization process, the membrane furthest carbon atom from the diffusion pore was kept frozen, while all other atoms were allowed to freely relax, to avoid membrane sheet drifting. Then, the molecule was gradually brought closer to the pore until the TS and the adsorbed state were reached. Minima and TS were characterized through frequency analysis, involving the construction and diagonalization of the Hessian matrix obtained by finite atomic displacements of 0.03 Å length.

The adsorption energy for O_2_ and N_2_ molecules is computed through the following expression:(1)$$E_{ads}^{i}=E_{S/i}-{(E}_{S}+E_{i})$$
where $E_{S/i}$ represents the energy with the *i*^th^ species adsorbed on the grazyne surface, $E_{S}$ is the energy of the pristine grazyne layer, and $E_{i}$ is the energy of the *i*^th^ species in vacuum and in their own ground state. Furthermore, the energy barrier for the diffusion across the membrane, $E_{b}^{i}$, is obtained through
(2)$$E_{b}^{i}=E_{TS,i}-E_{S/i}$$
where $E_{TS,i}$ is the energy of the TS for the *i*^th^ molecule diffusion across the grazyne pore. Both terms, $E_{ads}^{i}$ and $E_{b}^{i}$, were gained by adding the zero-point energy (ZPE) term, defined as
(3)$$E_{ZPE}=\frac{1}{2}\sum_{j}^{NMV}h\upsilon_{j}$$
where h is the Planck’s constant, and $\upsilon_{j}$ are the normal modes of vibration (NMV). The summation runs over all the NMV in the case of minima, whereas in the case of TS structures, the imaginary frequency is not included.

In addition, the rate constants associated to the diffusion process have been computed by means of the transition state theory (TST), according to the following expression:(4)$$r_{i}=\frac{k_{B}T}{h}\frac{q_{vib,i}^{\neq }}{q_{vib,i}^{ads}}e^{-\frac{E_{b}^{i}}{k_{B}T}}$$
where $k_{B}$ is the Boltzmann’s constant, *T* is the temperature, and $q_{vib,i}^{\neq }$ and $q_{vib,i}^{ads}$ represent the vibrational partition functions corresponding to the TS and the adsorption minimum, respectively. While TST is typically used in the context of chemical reactions, it is equally applicable to diffusion processes when interactions are made/broken, as TSs can be identified on any potential energy hypersurface, regardless of the type of process involved (molecular movement or chemical bond-breaking processes). Since both states imply a strong interaction between the molecule and the membrane, only the vibrational partition function is considered, as follows:(5)$$q_{vib,i}^{ads}=\prod_{j}^{NMV}\frac{1}{1-e^{-(\frac{hv_{j}}{k_{B}T})}}$$

Once the rates for each species are obtained, the selectivity of O_2_ relative to N_2_, denoted as $S_{O_{2}/N_{2}}$, can be determined according to,
(6)$$S_{O_{2}/N_{2}}=\frac{r_{O_{2}}}{r_{N_{2}}}$$

Furthermore, the description of desorption rates, $r_{des,i}$, in grazyne structures becomes possible. To achieve this, we applied the concept of a late TS within the framework of TST as follows:(7)$$r_{des,i}=S_{des,i}\nu_{des,i}e^{\frac{E_{ads}^{i}}{k_{B}T}}$$
where $S_{des,i}$ is the desorption coefficient, with values ranging from 0 to 1, representing the probability of an adsorbed molecule being desorbed. In this study, a desorption coefficient of one was assumed for both species. The term $\nu_{des,i}$ is defined as follows:(8)$$\nu_{des,i}=\frac{k_{B}T}{h}\frac{q_{trans,i}^{gas}q_{rot,i}^{gas}q_{vib,i}^{gas}}{q_{vib,i}^{ads}}$$
where $q_{trans,i}^{gas}$, $q_{rot,i}^{gas}$, and $q_{vib,i}^{gas}$ represent the translational, rotational, and vibrational partition functions, respectively, for the *i*^th^ species in the gas phase. The term $q_{vib,i}^{ads}$ was previously introduced and refers to the vibrational partition function of the same species in the adsorbed state.

Finally, the rate for a non-activated adsorption process can be determined using the Hertz–Knudsen equation
(9)$$r_{ads,i}=\frac{S_{0,i}p_{i}A}{\sqrt{2\pi m_{i}k_{B}T}}$$
in which $S_{0,i}$ represents the sticking coefficient that quantifies the probability that a molecule, upon reaching the membrane, will remain adsorbed. The terms $p_{i},$ $m_{i}$, and A denote the partial pressure, molecular mass, and adsorption area associated with N_2_ and O_2_, respectively. We define A as *a·b*, where these parameters are reflected in Table 1. The used sticking coefficient value is the conservative value of 0.2, consistent with values reported in similar studies [25]. The vibrational frequencies of the adsorbed N_2_ and O_2_ molecules and their TSs on the grazyne membranes, used in the rate calculations, are provided in Appendix A. These data include qualitative descriptions of the associated vibrational modes.

Additionally, classical MD simulations were carried out using the large-scale atomic/molecular massively parallel simulator (LAMMPS) package [26]. The simulation boxes used were 61.272 × 148.584 × 140 Å^3^ for the [1],[2]{2}-grazyne and 81.696 × 148.584 × 140 Å^3^ for the [1],[2]{(0,0),2}-grazyne. The grazyne membranes were located at half the *z* axis length along the *xy* plane, creating two sections with equal volumes. The adaptive intermolecular reactive empirical bond-order (AIREBO) potential was employed to define the interactions of the grazyne layer [27], thanks to its good representation of interatomic interactions within materials and compounds, predominantly constituted of carbon and hydrogen [28,29]. In the case of gas molecules, the intermolecular interactions were modelled by Lennard-Jones 12-6 potential plus a Coulomb interaction term [30], as follows:(10)$$V_{non-bonded}\left(r_{ij}\right)=\frac{q_{i}q_{j}}{4\pi \varepsilon_{0}r_{ij}}+4\varepsilon_{ij}((\sigma_{ij}/r_{ij}{)}^{12}-(\sigma_{ij}/r_{ij}{)}^{6})$$
where $\varepsilon_{0}$ is the vacuum permittivity, $\varepsilon_{ij}$ and $\sigma_{ij}$ are the Lennard-Jones parameters of the *ij* interaction, $r_{ij}$ is the interatomic distance between pairs, and $q_{i}$ and $q_{j}$ denote the partial atomic charges of the *ij* pair species. All the values used are listed in Table 2 and correspond to the transferable potentials phase equilibria (TraPPE) force field for the oxygen [31], which establishes a 1.210 Å bond length between oxygens, accompanied by a massless dummy atom positioned at the molecular center of mass. A charge of −0.113 *e* was assigned to each of the two oxygen atoms, balanced by the charge assigned to the dummy atom of 0.226 *e*. The N_2_ molecules were modeled based on the methodology proposed by Murthy et al. [32], with a bond length of 1.098 Å. In order to account for the experimental quadrupole moment inherent to the molecule, a massless dummy atom was placed at the center of mass with a charge of 0.964 *e*, thereby equalizing the charges of −0.482 *e* assigned to each nitrogen. The CO_2_ molecule was modeled according to the elementary physical model (EPM2) [33], in which the oxygen atoms are located at a bond length of 1.149 Å with respect to the carbon atom, resulting in a linear molecular geometry. The charge assigned to the C atom is 0.6512 *e* while on each oxygen atom the charge corresponds to −0.3256 *e*. Finally, Argon atoms have been modeled as a single particle with zero charge [30]. The unlike pair parameters were calculated using the arithmetic Lorentz–Berthelot combination rules. During the MD simulations, the molecules were treated as rigid entities.

All MD simulations were conducted under the canonical *NVT* ensemble, where the number of particles, *N*, the volume, *V*, and the temperature, *T*, of the system were fixed. Different values of pressure were studied to assess its significance in the process. To control the temperature of the system and keep it constant throughout the simulation, the Nose–Hoover thermostat was employed [34,35,36]. Initially, the gas mixture was thermalized for 100 ps, a process carried out avoiding gas–membrane interaction. To do so, the gas molecules are confined between two rigid walls, one directly above the grazyne membrane and another one at a distance ranging from 20 Å to 50 Å from the grazyne, depending on the desired pressure. Once the desired temperature was reached, the lower wall was removed and relocated at the bottom edge of the simulation box, then the diffusion process was started and was sustained over 500 ps, time that constitutes the production stage.

The number of molecules that diffuse across the surface within the production stage is commonly used to compute the membrane permeability, $P_{i}$, defined as [14],
(11)$$P_{i}=\frac{N_{i}}{A\cdot t}$$
where $N_{i}$ are the moles of net permeated gas molecules, A corresponds to the membrane area, and t corresponds to the simulation time.

## 3. Results

### 3.1. DFT Results

Before presenting the results obtained from DFT calculations, a series of considerations are needed. First, to determine whether a molecule is capable of passing through the pore, it is desirable to obtain values below 1 eV for diffusion energy barriers. In this sense, previous studies have determined that values below 0.3 eV would be adequate to ensure that molecules diffuse through these materials [19]. Furthermore, it is necessary to obtain moderate $E_{\mathrm{ads}}^{i}$ values, as low adsorption energies would result in insufficient gas attraction, whereas high adsorption energies could lead to chemisorption processes, deviating from the desired physisorption mechanism. Chemisorption would imply that the molecules could remain captured on the grazyne membrane surface instead of permeating through, making grazyne membranes effectively scrubber materials, and to act as membranes, it would be necessary to find suitable techniques to regenerate and clean grazynes from these adsorbed molecules.

Moreover, when investigating single molecules diffusion, scenarios involving the simultaneous presence of two molecules in the same pore are excluded, given their low probability, and the repulsive interactions they would have, frustrating their diffusion through the grazyne structure. Consequently, single passage of O_2_ and N_2_ molecules across grazyne membranes was studied yet considering two possible molecular orientations for each molecule type, either with the molecular axis being perpendicular or parallel to the grazyne plane. The obtained energetic values are listed in Table 3. For both structures, the adsorption of O_2_ molecules is stronger than that of N_2_, regardless of the considered orientation. Another positive income comes from the fact that both species physisorb ensuring no blockage of the membrane. In addition, diffusion O_2_ barriers are totally non-sensitive to the molecular orientation whereas N_2_ slightly prefers the perpendicular path. According to the energetic values reported in Table 3, the [1],[2]{(0,0),2}-grazyne membrane exhibits lower diffusion barriers than [1],[2]{2}-grazyne.

Using the values from Table 3, the energy profile for the diffusion of the studied molecules has been represented in Figure 3. It is evident that O_2_ experiences lower energy barriers for both conformations with respect to the N_2_ cases. Among the studied conformations, the perpendicular orientation appears to be more suited to carry out the diffusion through the grazyne membranes for both molecules.

The TS visual inspection helps at identifying different situations. For instance, TS (*cf.* Figure 4) for N_2_ is obtained when the adsorbate is positioned at 0 Å from the pore center, with the layer in a completely flat conformation. In the case of O_2_, the TS corresponds to the molecule slightly tilted from the completely perpendicular orientation in both structures.

The adsorbed states of the molecules also appear to be distinct. For O_2_, grazyne bulges toward the molecule, indicating an attraction between the two parts. Conversely, for N_2_ the same grazyne bulges away from the molecule, resulting in a repulsive interaction, as observed in Figure 5. Apparently, the energy differences shown in Table 3 could be attributed to these structural discrepancies obtained in the TS and adsorbed state of each molecule. The planar configuration observed in the TS and the bulged structure in the adsorbed state are not a result of constraints applied during the simulations. Instead, these geometries arise naturally as the system evolves to minimize its energy. This behavior reflects the response of the grazyne membranes to the interaction with gas molecules under the simulated conditions.

To further analyze the factors affecting the diffusion of N_2_ and O_2_ through the grazyne membranes, the electrostatic potential maps for both grazyne structures were acquired and shown in Figure 6. There, it is evident that the electrostatic potential is uniformly distributed around the atoms of the material, and nearly zero within the pore region, implying that the diffusion is not biased by the electrostatic potential, yet in turn, primarily governed by the pore size and the size of the molecule.

### 3.2. Rate Constants and Selectivity

According to the energy barriers obtained in the previous section, O_2_ should pass more easily than N_2_ on both grazyne membranes. In order to further verify this hypothesis, rate constants have been calculated for each species at different temperatures by means of Equation (4) and represented in Figure 7. For each rate constant the two orientations studied—perpendicular and parallel—have been considered. Clearly, O_2_ exhibits a higher diffusion in both grazynes, reaching values ca. 12 s^−1^ at 500 K. For both O_2_ and N_2_, the [1],[2]{2}-grazyne yields the largest rate constant values, even though the difference with the values obtained in [1],[2]{(0,0),2}-grazyne is very small. Furthermore, with an increase in temperature, the rate constant also rises, as expected. The temperature trend observed can be explained from a physical perspective, where higher temperatures provide molecules with greater kinetic energy and velocity, thereby facilitating the diffusion across the pores. According to the different rate constants, it is reasonable to consider both membranes for O_2_/N_2_ separation and identify the appropriate range of temperatures and pressures that enhance the separation.

Figure 8 represents the selectivity of both membranes towards O_2_ as a function of temperature. The results show that, as the temperature increases, both membranes become less selective. Thus, at 150 K, the selectivity value is 3.5 and 3.2 for [1],[2]{(0,0),2}-grazyne and [1],[2]{2}-grazyne, respectively. However, at 300 K, both grazyne membranes achieve a selectivity value of 1.8, indicating that approximately twice as much O_2_ would be filtered from the mixture, ensuring the capability of purifying a mixture of these gases even at room temperature.

### 3.3. Kinetic Phase Diagrams

The analysis of the ability of materials to adsorb molecules is essential in designing effective filtration systems. The effectiveness of a material in this regard depends on its ability to adsorb target molecules upon, and if so, in the regeneration for a continued use. To evaluate this process, we created Kinetic Phase Diagrams (KPDs) that illustrate the adsorption and desorption tendency of N_2_ and O_2_ on both grazynes as a function of the temperature and pressure of the system [37,38]. Under equilibrium conditions, the rates of adsorption and desorption are equal. This equilibrium prompts the search for specific temperatures and pressures that achieve this balance. Figure 9 clearly shows two different regions; one where molecular adsorption is preferred, and another region where desorption is preferred, separated by the equilibrium line. Keeping that in mind, it is worth noting that O_2_ exhibits a superior adsorption range on both types of grazynes, consistent with the stronger $E_{\mathrm{ads}}^{i}$ values reported in Table 3, *vide supra*.

The [1],[2]{(0,0),2}-grazyne displays lower adsorption range, positioning it as a slightly more favorable option for operating at high pressure and temperature conditions. However, what is most significant is that under standard conditions (*p* = 1 atm and *T* = 300 K), neither of the molecular species exhibit a propensity for adsorption, making these grazynes potential candidates for their use as separation membranes, as they would not, in principle, get their pores poisoned by gas adsorption.

### 3.4. Molecular Dynamics

Molecular dynamics (MD) simulations allow a direct comparison with results obtained by DFT calculations and check whether both methodologies are in line and complement each other. While MD focuses on capturing the dynamic aspects of processes, DFT has so far been limited to addressing thermodynamics and static kinetic approximations. The results of these MD simulations—the number of molecules of each species crossing the membrane—have been used to calculate the permeability of grazynes from Equation (11). Here, two types of simulations can be differentiated depending on the gas mixture that has been used. In the first type, a gas mixture with equimolar composition of O_2_ and N_2_ has been modeled (i.e., 64 molecules of O_2_ and 64 molecules of N_2_) and the diffusion across both membranes was evaluated at different pressures. For the first set of simulations, three different simulations were performed for each pressure, with the variation among replicates arising from differences in their initial states. These additional simulations serve as independent replicates, enabling assessment of the robustness of the results and confirming that the observed trends are not merely due to probabilistic situations. The results of these simulations are shown in Table 4 and Table 5. Here, O_2_ exhibited a greater capacity for crossing the membrane, a fact that is consistent with a lower DFT energy barrier associated with O_2_ (*cf.* Table 3). To be exact, approximately twice as many O_2_ molecules were filtered in [1],[2]{2}-grazyne. In the [1],[2]{(0,0),2}-grazyne, the difference is even more noticeable where O_2_ represented up to *ca.* 95% of the total filtered gas species, which implies an increase of *ca*. 45%. These differences are maintained for all pressure values, where a small tendency to decrease the number of filtered molecules with decreasing pressure can be seen. In fact, the values reported in Table 4 show that by decreasing the operative pressure, the membrane generates a purer O_2_ phase using the [1],[2]{2}-grazyne. Note that the pressure indicated for each simulation represents only the initial pressure of the gas on the feeding side. As seen in the MD (see Appendix A), the deformation and vibration of the membrane over time slightly modifies the cell volume and shape at each timestep, making it challenging to accurately evaluate the feed pressure throughout the simulation. When one focuses on the replicates, for each pressure, oscillations are observed in the number of filtered molecules. However, a clear trend is observed with O₂ consistently exhibiting the highest filtration rate.

Tracking the molecules that come across the membrane at each timestep enables the generation of Figure 10, where the number of molecules in the permeated side is represented over time. The thermalization process, occurring from 0 to 100 ps, is omitted. Initially, both molecules exhibit a comparable diffusion rate. However, before reaching 200 ps, there is a noticeable shift in trends. Nitrogen diffusion appears to decelerate, whereas oxygen maintains its initial diffusion pattern unaltered.

By using the data reported in Table 4, the permeability was computed and represented in Figure 11. Comparing both grazynes, it becomes evident that [1],[2]{2}-grazyne exhibits higher permeability values for both molecules, thus being a less restrictive membrane to O_2_/N_2_ diffusion. Furthermore, O_2_ molecules displayed elevated permeability values across both grazynes, underscoring their superior diffusion capacity.

### 3.5. O_2_ Direct Air Capture

Considering the results obtained through DFT and MD analyses, it appears plausible to achieve the separation of these molecules using the studied grazynes, thanks to their different diffusion rates and permeability, with oxygen showing greater ease of passage through the membranes. However, the study carried out up to this point only considers N_2_ and O_2_ molecules, when there are more compounds in the air that can affect the separation. To have a more representative view regarding the separation of these molecules, a new set of MD simulations were carried out considering, apart from N_2_ and O_2_, CO_2_ molecules and Ar atoms—the main compounds after N_2_ and O_2_ in air—plus the mixture was made so that the composition was as close as possible to that of atmospheric dry air. Table 6 shows the number of molecules and their contribution to the four components initial mixture.

The size of the simulation box was expanded to accommodate all the molecules, being of 152.524 × 373.576 × 2996.384 Å^3^ for the [1],[2]{2}-grazyne and 204.240 × 373.575 × 2996.384 Å^3^ for the [1],[2]{(0,0),2}-grazyne.

Table 7 brings together the results obtained in these simulations. The trend observed in the previous simulations is replicated also here, where the filtered gas was notably enriched in O_2_. Particularly noteworthy is the result for the [1],[2]{(0,0),2}-grazyne, where oxygen increases its presence up to 47% when it only constituted 21% in the initial mixture. On the other hand, N2 presence is reduced to 60% and 51% in the [1],[2]{2}- and [1],[2]{(0,0),2}-grazyne, respectively. However, a downside of these simulations is the permeation of CO_2_, which, despite starting with only four initial molecules, managed to penetrate the [1],[2]{2} membrane, as anticipated based on previous findings where CO_2_ diffusion was documented [19,20]. Moreover, the quantity of filtered molecules is greatly influenced by their interaction with the membrane, particularly when the approach occurs perpendicular or parallel to its surface. This fact underscores the significance of understanding the molecular dynamics and structural characteristics of the membrane to predict and optimize separation processes effectively. The [1],[2]{(0,0),2}-grazyne remained impermeable to CO_2_ initially, though with prolonged simulation, some CO_2_ molecules might penetrate this membrane, akin to the other model. While argon atoms also crossed the membranes, their inert nature renders them non-threatening. From a broader perspective, [1],[2]{2}-grazyne allows a greater number of total molecules to pass compared to its counterpart, indicating its less restrictive nature. Specifically, 932 molecules were filtered in [1],[2]{2}, whereas only 452 were filtered in [1],[2]{(0,0)2}. Despite this, the gas filtered through [1],[2]{2} is less enriched in O_2_, representing only 37%. According to these results, the effectiveness of both membranes to generate richer O_2_ mixtures directly from N_2_/O_2_ and dry air mixtures is notorious.

In this work, we focused on anhydrous conditions for the gas separation study, as dry gases are commonly used in gas separation tests. Consequently, the effects of moisture or air vapors were not explicitly considered. However, we acknowledge the importance of considering these factors, as seen, e.g., on water on the adsorption and diffusion processes through graphene [39]. Future advances in the present work should thus consider the influence of moisture, vapors, and various impurities on the performance of grazyne-based membranes, providing a more comprehensive understanding of their behavior under more realistic conditions.

## 4. Conclusions

The present DFT and MD calculations analyzing N_2_ and O_2_ permeation through nano-engineered grazyne membranes in a holistic fashion, considering thermodynamic, kinetic, and dynamic aspects yield consistent results, indicating the potential use of grazynes to separate N_2_ and O_2_ from air. DFT calculations show the physisorption of both molecules on the grazyne membrane models, preventing the pore obstruction by chemical adsorption, further confirmed by KPD under working temperature and pressure conditions. Additionally, the DFT calculations suggest that the two types of molecules could permeate the membrane in two different orientations: perpendicular and parallel. In this sense, O_2_ exhibited a significantly higher diffusion capacity compared to N_2_, attributed to its smaller diffusion barriers. This trend is reflected in the diffusion rates where O_2_ obtained systematically higher rates than N_2_ but with both rates higher than 1 s^−1^. Despite the significant differences in the values, working at low temperatures would be preferable to enhance selectivity towards O_2_. Finally, the electrostatic potential maps indicate that the diffusion of N_2_ and O_2_ through the grazyne membranes is primarily influenced by the pore size and the size of the molecule rather than by electrostatic interactions between grazyne and molecules’ electron densities.

MD simulations with equimolar mixtures of O_2_:N_2_ clearly indicate a selectivity towards O_2_ in both grazynes. To be precise, approximately twice as many O_2_ molecules were filtered in [1],[2]{2}-grazyne whereas in the [1],[2]{(0,0),2}-grazyne, the difference is even more pronounced, with O_2_ representing up to ca. 95% of the filtered gas, implying an increase of ca. 45%. These data align perfectly with the selectivity obtained through DFT calculations, where a selectivity towards O_2_ close to 2 at 300 K was observed. The simulations also show promising results in the case of dry air. Here, the trend observed for the O_2_/N_2_ mixture was confirmed, with O_2_ being enriched in the permeate gas for both grazynes. Particularly noteworthy is the result for the [1],[2]{(0,0),2}-grazyne, where oxygen increases its presence by 26%, nearly constituting half of the filtered gas (i.e., 47%), despite only constituting 21% of the initial gas. On the other hand, [1],[2]{2}-grazyne allows a greater number of molecules to diffuse across the membrane, indicating its less restrictive nature. However, despite this difference, the gas filtered through [1],[2]{2} is less enriched in O_2_, representing only 37%. Nonetheless, the objective of obtaining enriched oxygen mixtures remains achievable.

All in all, the present study, which explores the energetics, kinetics, and dynamics of O_2_ separation from N_2_ using engineered grazyne membranes, paves the way for their application. However, the results underscore that it is essential to control the operating conditions to maximize the separation efficiency. Further research may be necessary to understand the separation behavior and consider the possible impact of other species present in untreated O_2_/N_2_ streams.

## Figures and Tables

**Figure 1 nanomaterials-14-02053-f001:**
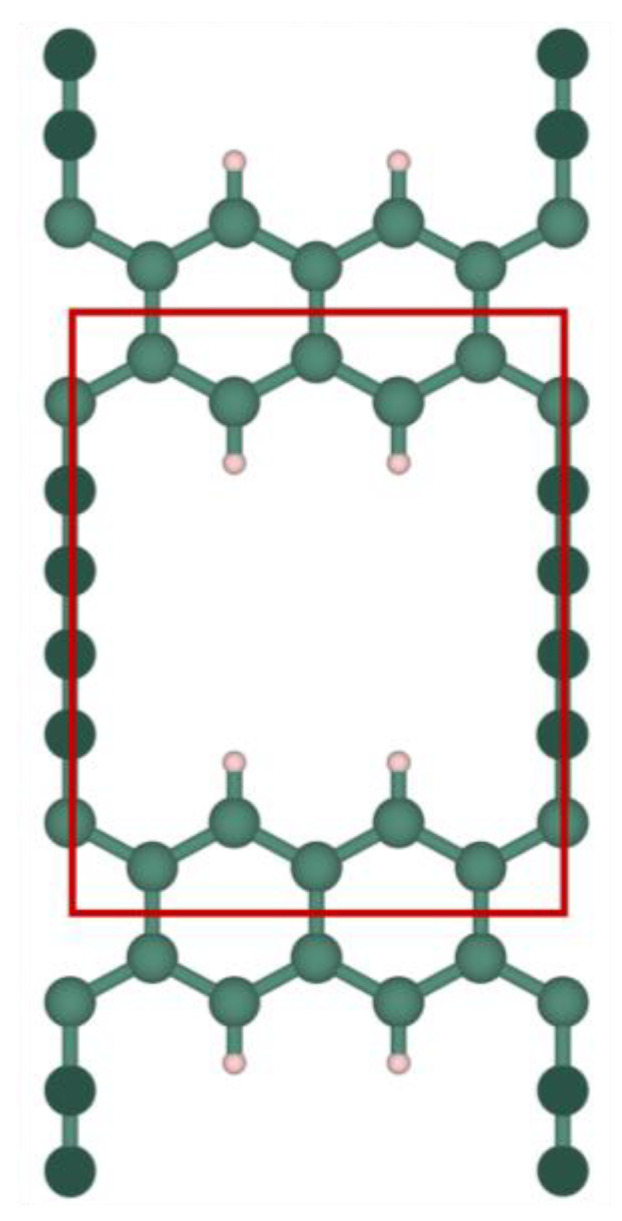
Top view of [1],[2]{2}-grazyne. Dark and light green spheres correspond to *sp-* and *sp*^2^-C atoms, respectively, while white spheres denote H atoms. The red lines fence the unit cell.

**Figure 2 nanomaterials-14-02053-f002:**
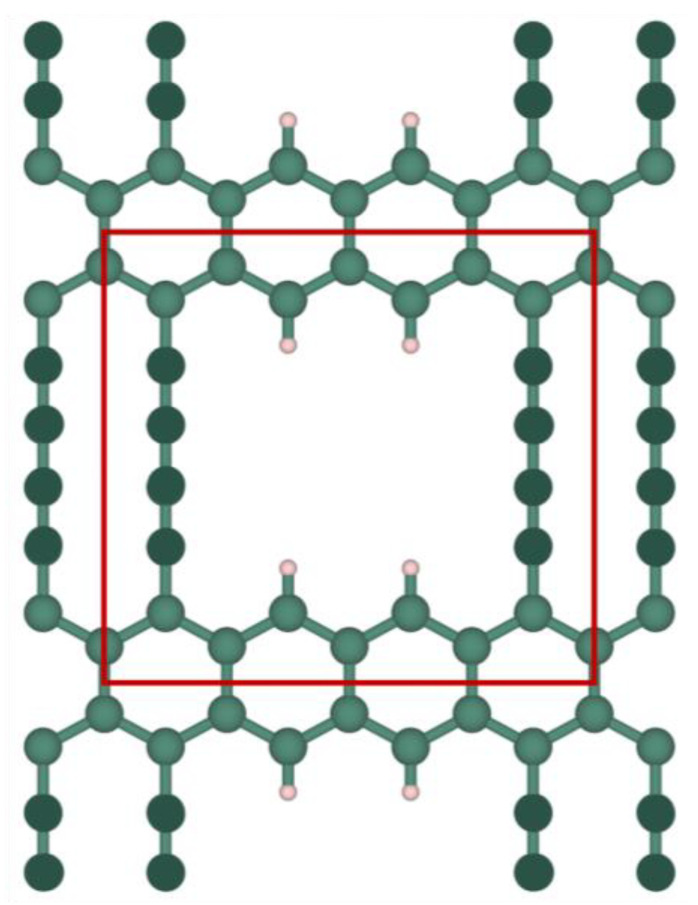
Top view of [1],[2]{(0,0),2}-grazyne. Color code as in Figure 1.

**Figure 3 nanomaterials-14-02053-f003:**
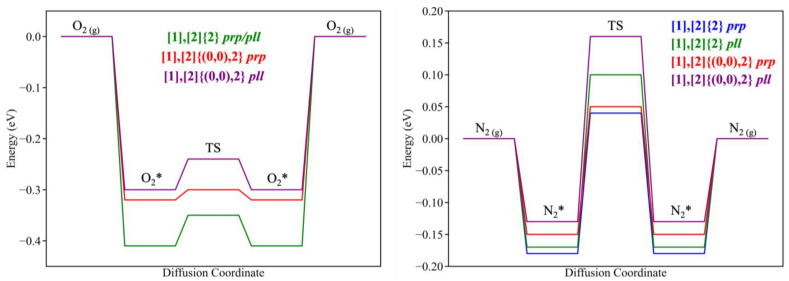
Diffusion paths for O_2_ (left) and N_2_ (right) across both grazyne membranes for the perpendicular (*prp*) and parallel (*pll*) conformations. TS represents the diffusion transition states and O_2_* and N_2_* the absorbed states of both molecules on the grazyne. Note that in the case of O_2_ and the [1],[2]{2}-grazyne, the perpendicular and parallel diffusion energy paths are essentially coincidental and cannot be distinguished.

**Figure 4 nanomaterials-14-02053-f004:**
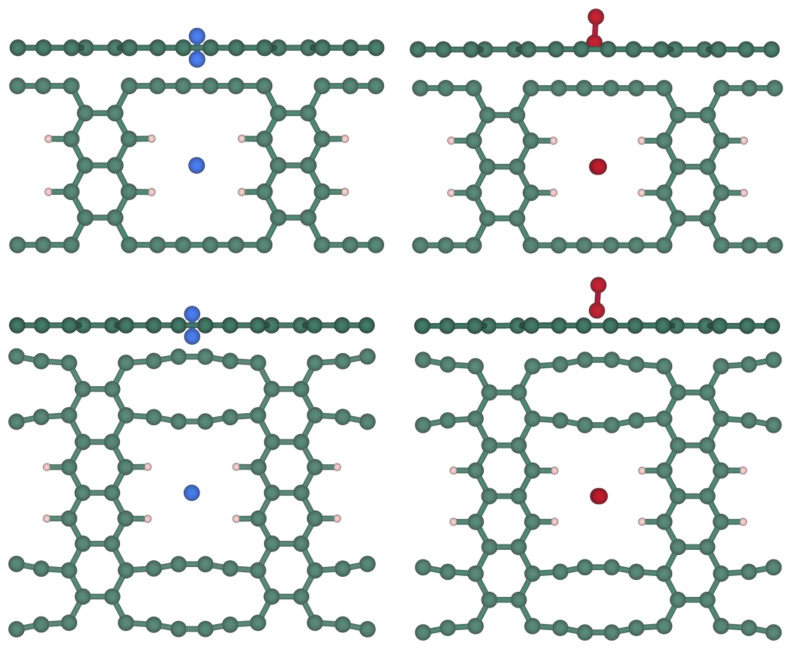
Side (**top**) and top (**bottom**) views of the TSs corresponding to the diffusion process of N_2_ (**left**) and O_2_ (**right**) across the [1],[2]{2}-grazyne (upper images) or the [1],[2]{(0,0),2}-grazyne (lower images). Blue spheres denote N atoms while red ones denote O atoms. The rest of the code coloring is as in Figure 1.

**Figure 5 nanomaterials-14-02053-f005:**
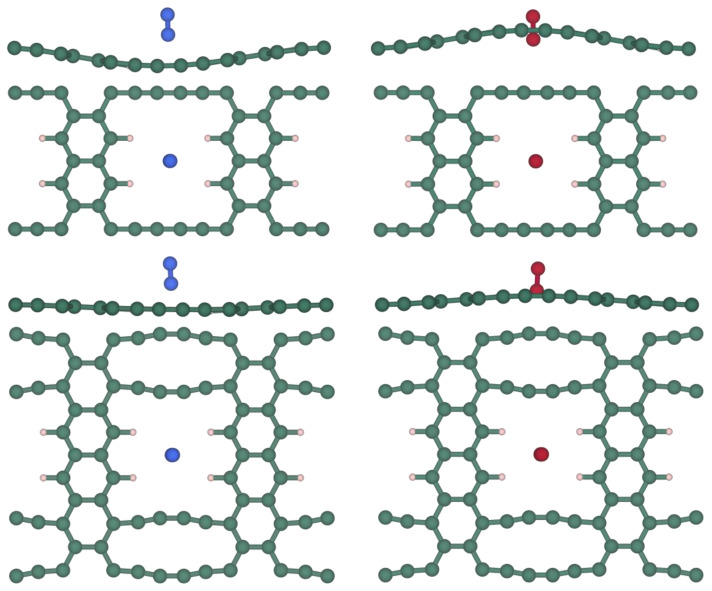
Side (**top**) and top (**bottom**) views of the N_2_ (**left**) and O_2_ (**right**) molecules adsorbed states on the [1],[2]{2}-grazyne (upper images) and the [1],[2]{(0,0),2}-grazyne (lower images). Coloring code as in Figure 2.

**Figure 6 nanomaterials-14-02053-f006:**
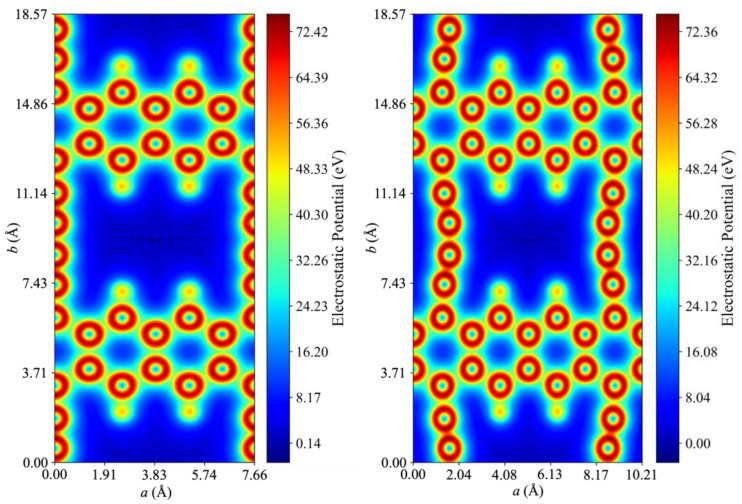
Electrostatic potential maps for [1],[2]{2}-grazyne (**left**) and [1],[2]{(0,0),2}-grazyne (**right**) pores.

**Figure 7 nanomaterials-14-02053-f007:**
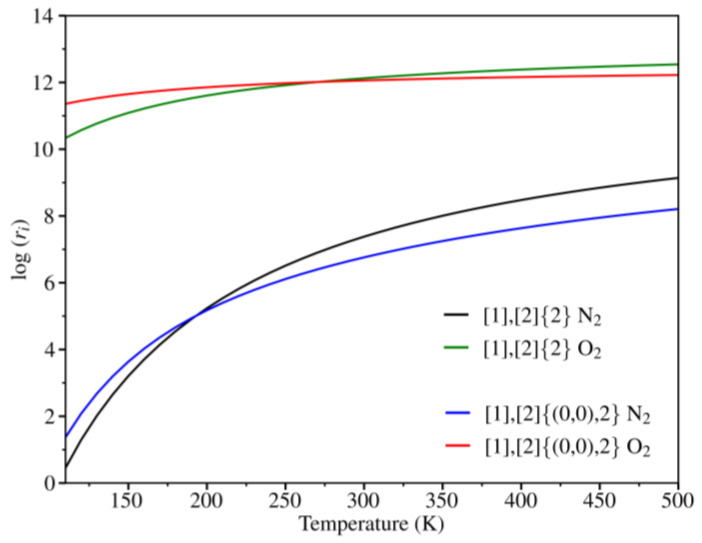
Rate constants, $r_{i}$, for O_2_ and N_2_ diffusion across the grazyne membranes as a function of temperature.

**Figure 8 nanomaterials-14-02053-f008:**
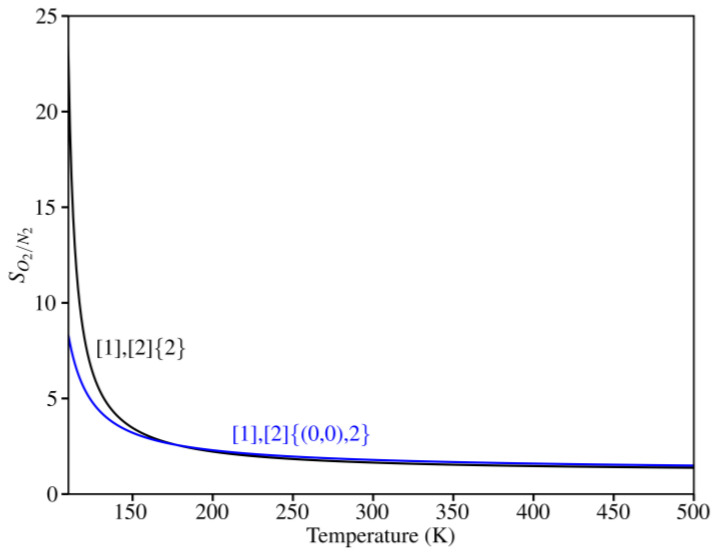
O_2_ selectivity over N_2_ for the studied grazynes in the range of 100–500 K.

**Figure 9 nanomaterials-14-02053-f009:**
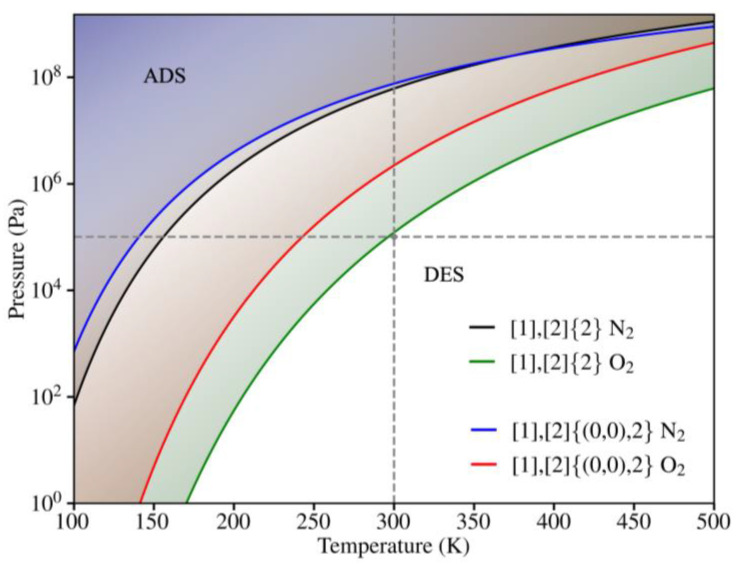
KPD of [1],[2]{2}-grazyne (black for N2 and green for O2) and [1],[2]{(0,0),2}-grazyne (blue for N_2_ and red for O_2_) as a function of gas pressure and temperature. White regions denote preference for desorption (DES), whereas faded ones go for adsorbed situations (ADS). The grey dotted line indicates a total pressure of *p* = 1 atm and *T* = 300 K.

**Figure 10 nanomaterials-14-02053-f010:**
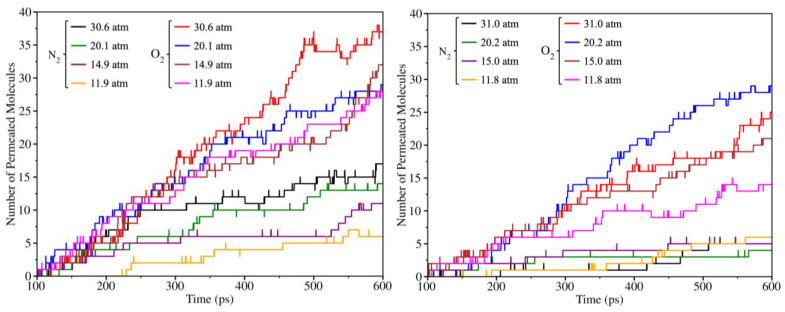
Number of O_2_ and N_2_ molecules permeated across [1],[2]{2}-grazyne (**left**) and [1],[2]{(0,0),2}-grazyne (**right**) membranes as a function of simulation time, at *T* = 300 K, and at different gas pressures.

**Figure 11 nanomaterials-14-02053-f011:**
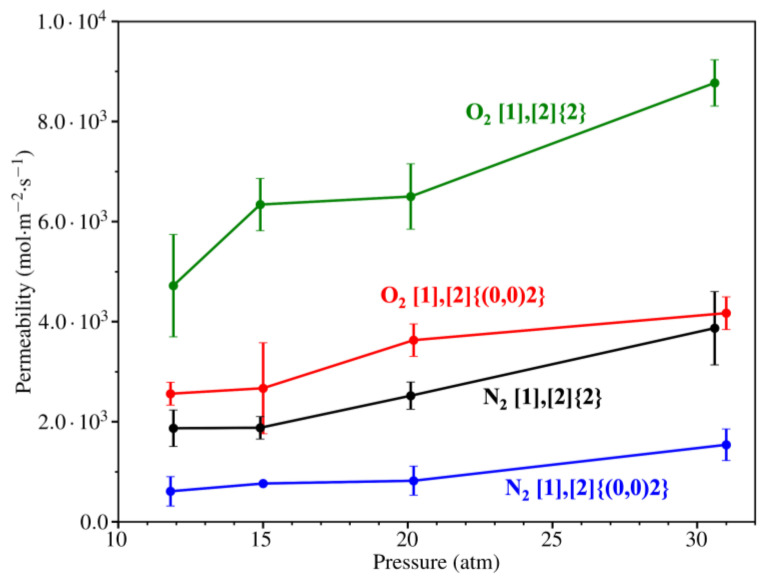
Permeability at different gas stream pressures for O_2_ or N_2_ on both studied grazyne membranes and at *T* = 300 K. The permeability values are calculated from the average of multiple simulation replicates, with associated error bars representing the variability of the results.

**Table 1 nanomaterials-14-02053-t001:** Optimized cell parameters, a and b, for [1],[2]{2}- and [1],[2]{(0,0),2}-grazyne membranes.

Grazyne	*a*/Å	*b*/Å
[1],[2]{2}	7.659	18.573
[1],[2]{(0,0),2}	10.212	18.573

**Table 2 nanomaterials-14-02053-t002:** Non-bonded parameters for O_2_, N_2_, CO_2_, and Ar.

Molecule	Atom, i	$q_{i}$/*e*	$\varepsilon_{i}\cdot k_{b}^{-1}/$eV	$\sigma_{i}$/Å
O_2_	O	−0.1130	49.000	3.020
	dummy	+0.2260	—	—
N_2_	N	−0.4820	36.400	3.320
	dummy	+0.9640	—	—
CO_2_	C	+0.6512	28.129	2.757
	O	−0.3256	80.507	3.033
Ar	Ar	0.0	119.800	3.405

**Table 3 nanomaterials-14-02053-t003:** Adsorption energies, $E_{ads}^{i}$, and diffusion energy barriers, $E_{b}^{i}$, for O_2_ and N_2_ on both grazyne structures, for perpendicular and parallel orientations. All values are given in eV.

Grazyne	Perpendicular				Parallel			
	$E_{\mathrm{ads}}^{O_{2}}$	$E_{b}^{O_{2}}$	$E_{\mathrm{ads}}^{N_{2}}$	$E_{b}^{N_{2}}$	$E_{\mathrm{ads}}^{O_{2}}$	$E_{b}^{O_{2}}$	$E_{\mathrm{ads}}^{N_{2}}$	$E_{b}^{N_{2}}$
[1],[2]{2}	−0.41	0.06	−0.18	0.22	−0.41	0.06	−0.17	0.27
[1],[2]{(0,0),2}	−0.32	0.02	−0.15	0.20	−0.30	0.02	−0.13	0.29

**Table 4 nanomaterials-14-02053-t004:** Number of permeated gas molecules in [1],[2]{2}-grazyne for an equimolar initial mixture of O_2_/N_2_ at *T* = 300 K and at different pressures, for the three different MD runs.

	[1],[2]{2}-grazyne											
P (atm)	30.6			20.1			14.9			11.9		
#R	1	2	3	1	2	3	1	2	3	1	2	3
#O_2_	37	40	34	28	30	32	32	24	29	28	14	18
#N_2_	17	25	21	14	14	13	11	13	13	6	9	9
%O_2_	68.5	61.5	61.8	66.7	68.2	71.1	74.4	64.9	69.0	82.4	60.9	66.7

**Table 5 nanomaterials-14-02053-t005:** Number of permeated gas molecules in [1],[2]{(0,0),2}-grazyne for an equimolar initial mixture of O_2_/N_2_ at *T* = 300 K and at different pressures, for the three different MD runs.

	[1],[2]{(0,0),2}-grazyne											
P (atm)	31.0			20.2			15.0			11.8		
#R	1	2	3	1	2	3	1	2	3	1	2	3
#O_2_	25	33	20	29	26	22	21	13	13	14	15	20
#N_2_	5	12	12	4	9	5	5	5	4	6	2	1
%O_2_	83.3	73.3	62.6	87.8	74.3	81.5	80.8	72.2	76.5	70.0	88.2	95.2

**Table 6 nanomaterials-14-02053-t006:** Gas mixture composition used in the MD simulations.

Molecule Type	Number of Molecules	%
N_2_	7800	77.97
O_2_	2100	20.99
Ar	100	1.00
CO_2_	4	0.04

**Table 7 nanomaterials-14-02053-t007:** Number of molecules filtered through the grazyne membranes for a N_2_/O_2_/Ar/CO_2_ mixture.

	[1],[2]{2}-grazyne		[1],[2]{(0,0),2}-grazyne	
	#Permeated Molecules	%Filtered Gas	#Permeated Molecules	%Filtered Gas
N_2_	562	60.30	231	51.11
O_2_	351	37.66	213	47.12
Ar	18	1.93	8	1.77
CO_2_	1	0.11	0	0.00

## Data Availability

Data available upon request from the authors.

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
