# Peer review of "Selective O2/N2 Separation Using Grazyne Membranes: A Computational Approach Combining Density Functional Theory and Molecular Dynamics"

_nanomaterials, 2024, doi:10.3390/nano14242053_

Round 1
Reviewer 1 Report
Comments and Suggestions for Authors
Using DFT and classical molecular dynamics simulations, the authors study the separation of N2 and O2 from the gas phase mixture by two grazyne porous membranes. They determine the binding energies and transition state configurations and energies for the passing of the molecules through the membrane pores. They also determine the rate of passing of the molecules through the pores using assumptions from transition state theory. The results of the classical MD are shown to be consistent with the DFT calculations.
This manuscript is well written and clearly presented. It includes an excellent review of previous work on separating N2 / O2. The computational protocols are suitable for these systems. The analysis of the results are also valid. I believe the analysis can be somewhat extended to make the work more impactful and suggest some items below. If some of these are implemented, I can recommend the manuscript for publication in Nanomaterials.
-Do the authors have the potential energy profile for the N2 / O2 species as a function of the perpendicular distance from the grazyne plane? This would be an interesting complement to the value of the binding and TS energies.
-From the VASP calculations, can the authors determine an electrostatic potential map of the pore area to determine how much this factor would affect the N2 / O2 molecules?
-Can rough error bars be places on the permeability values shown in Fig. 9? The trends in Fig. 8 from which Fig. 9 was determined seem to have considerable fluctuations.
-Is there any study on the effect of moisture / air vapors on the operation of the membrane with respect to adsorption and effect on N2 / O2 diffusion? This point should at least be discussed.
-It would give the reader some intuition of the process if an animation of the MD results for the two grazynes could be prepared. The different types of behaviors of the molecules would be seen from this. These can be added as Supporting Information.
-Can the authors provide further details (qualitative and quantitative) regarding the nature of the vibrational modes of the adsorbed N2 / O2 molecules on the grazyne surfaces which were used to calculate the rates? These can be placed in a Supporting Information section.
Reviewer 2 Report
Comments and Suggestions for Authors
Please see attached review comments.

Round 2
Reviewer 1 Report
Comments and Suggestions for Authors
I would like to thank the authors for addressing my comments. The context and content of the manuscript have become clearer and I recommend it's publication in Nanomaterials.
Reviewer 2 Report
Comments and Suggestions for Authors
The authors have thoroughly addressed all the specific points raised, and the revised manuscript has been significantly improved as a result. Based on these revisions, the current manuscript is suitable for acceptance.